# MoXCo: How I learned to stop exploring and love my local minima?

Esha Singh[1], Shoham Sabach[2], Yu-Xiang Wang[3]
[1]CSE, UC San Diego, [2] Technion – Israel Institute of Technology
[3] Halıcıoğlu Data Science Institute, UC San Diego
{e3singh, yuxiangw}@ucsd.edu, ssabach@gmail.com

Deep neural networks are well-known for their generalization capabilities, largely attributed to optimizers' ability to find "good" solutions in high-dimensional loss landscapes. This work aims to deepen the understanding of optimization specifically through the lens of loss landscapes. We propose a generalized framework for adaptive optimization that favors convergence to these "good" solutions. Our approach shifts the optimization paradigm from merely finding solutions quickly to discovering solutions that generalize well, establishing a careful balance between optimization efficiency and model generalization. We empirically validate our claims using two-layer, fully connected neural network with ReLU activation and demonstrate practical applicability through binary quantization of ResNets. Our numerical results demonstrate that these adaptive optimizers facilitate exploration leading to faster convergence speeds and narrow the generalization gap between stochastic gradient descent and other adaptive methods.

## 1. Introduction

Vapnik [1999], Vapnik and Chervonenkis [1982] demonstrated that if any problem is learnable, it can also be learned by solving Empirical Risk Minimization (ERM). This insight has influenced generations of researchers, leading them to treat the statistical problem of *generalization* and the computational problem of *optimization*, i.e., solving REM, as entirely separate issues.

However, in modern deep learning models, the boundaries between optimization and generalization have become less distinct. For classification tasks, numerous solutions can achieve zero-error (and nearly zero-loss), yet they generalize very differently [zhang et al., 2021]. The outcome is intricately dependent on factors such as the initial model configuration, the choice of optimization algorithm, hyperparameter settings, and even an element of randomness.

This contrast is even more pronounced in regression tasks, where striving for the global optimal solution in Empirical Risk Minimization (ERM) may not be the most appropriate objective. For instance, if the data satisfies $y_i = f_0(x_i) + \mathcal{N}(0, \sigma^2)$ for $i = 1, ..., n$, a solution that achieves a zero square loss w.r.t. $y_1, \ldots, y_n$ is clearly an overfitting solution (except in a very specialized regime known as benign overfitting [Bartlett et al., 2020]). It was recently proven that benign overfitting does not happen ReLU networks for regression tasks [Joshi et al., 2023]. A better solution of interest should be one that is close to $f_0$ and has an expected error of $\sigma^2$. In other words, our primary goal should no longer be optimizing for the sake of minimizing a given objective function as much as possible. Instead, we should think about how to design optimization algorithms that converge to good local stable regions (i.e. flat local minima) that generalize better.

The energy landscape in modern deep learning is remarkably intricate. There are several pitfalls that an optimizer such as SGD can get stuck into or explode from such as plateaus and cliffs, sharp local minima, flat stationary points etc. This motivated many strategies in training DL models, spanning architectural designs like residual connections and batch normalization, as well as optimization tricks such as ADAM, gradient clipping, and learning rate scheduling etc. While these methods help navigate these pitfalls and promote exploration, recent work Malviya et al. [2023], Dehghani and Samet [2020] demonstrate that exploration alone is insufficient for finding optimal solutions.

Second Conference on Parsimony and Learning (CPAL 2025).

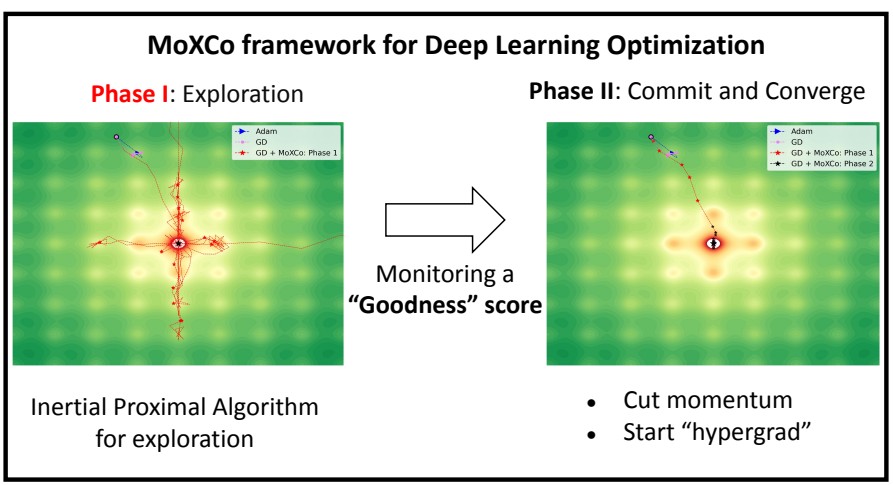

Figure 1: Optimization trajectories on Ackley function with multiple local minima: MoXCo (red), ADAM (blue), and GD (pink). Starting from top-left, MoXCo achieves global minimum through two phases: initial aggressive exploration (red dots) followed by consistent convergence in estimated vicinity (black dots). ADAM and GD quickly converge to local minima. Loss plot in Fig 9 demonstrates MoXCo's faster convergence speed. Note: In all loss contours lighter color corresponds to smaller loss values.

This observation motivates our key insight: while exploration can be made more aggressive to better traverse the loss landscape, an effective optimizer must also recognize when to transition from exploration to precise convergence upon detecting proximity to a promising local optima.

## 1.1. Summary of results

In this paper, we build on this idea and propose a new framework - **Mo**mentum e**X**ploration and **Co**mmit (MoXCo) for making such determination. MoXCo operates in two phases: an agressive exploration phase driven by momentum-based acceleration in black-box first-order optimization, followed by a strategic exploitation phase. The transition between phases is governed by a *goodness score* that quantifies the optimization potential of local regions. When this score exceeds a theoretically-motivated threshold, indicating the proximity to a favorable optimum, we initiate precise convergence by reducing momentum. This dynamic transition enables MoXCo to effectively balance aggressive exploration with targeted exploitation of favorable optima.

The key research questions that underlines the design of MoXCo are:

1. How to navigate any loss landscape to bypass underfitting local and overfitting global minima?
2. Can we identify and quantify the proximity of such "good" solutions, if present?
3. Additionally, do these solutions enhance performance, and can existing methods be utilized to address these challenges?

Below, we summarize our main observations and findings that help answer these questions.

**Why aggressive exploration is required.** Section 3 introduces momentum equations that promote exploration in optimization algorithms. Our empirical analysis shows that during early optimization steps, promoting exploration yields better results than rapid convergence, regardless of initialization. While both larger learning rates Lewkowycz et al. [2020] and momentum can facilitate exploration, we focus on momentum-based methods and demonstrate that adding a 'second' momentum term (Eq 2) enables more aggressive exploration of the loss landscape that reach to flatter minima. Section 4(i) provides empirical evidence supporting the effectiveness of this additional momentum component.

**Why committing-and-stopping is helpful.** While aggressive exploration is desirable especially in complex loss landscapes, is it alone sufficient to attain optimal solutions? The answer to this

question leads to second key component in the MoXCo framework (Section 3.2): a "commit-to-stop" mechanism, activated when one is in an ideal vicinity. This ideal locality is quantified by our *Goodness Score*, characterizing local geometrical properties as formalized in Theorem 1. The phenomena of commit-to-stop not only helps in deciding when to commit but also prevents overshooting or oscillating issues which may arise due to unpredictable interactions between learning rate and momentum hyper-parameters when they are improperly tuned. Thus, goodness score allows also helps to reduce the undesirable effects stemming from hyper-parameter sensitivity of adaptive methods & it's SGD variants Sivaprasad et al.. Section 5.1 shows numerical results that helps us illustrate these claims. Dehghani and Samet [2020] corroborates that exploration alone is insufficient.

**Altogether, do we reach *good* solutions?** While momentum-based methods are empirically observed to accelerate convergence speeds Goodfellow et al. [2016], theoretical understanding about its dynamics remains limited. Notably, their impact on generalization error suggests that adaptive optimizers, such as ADAM, generalize poorly in comparison to SGD. Additionally, it has been found that while momentum might speed up training, it can adversely affect generalization Wilson et al. [2017], Hardt et al. [2016]. We provide extensive experimental evidence (Section 4) to support our claims that our method consistently achieves better solutions.

## 1.2. Related Work

**Momentum methods** The two components of the MoXCo framework are elaborated in detail in sections 3.1, 3.2. The utilization of inertial force for momentum dates back to Polyak [1964]. Ochs et al. [2014] used this inertial force to support the use of a proximal gradient-type method for handling structurally induced regularization. Wang et al. [2023] uses inertial accelerated stochastic gradient methods to solve the low-rank CP decomposition problems. Similar to our second Eq.2, the estimation of stochastic gradients on a slightly perturbed point, for non-smooth & non-convex problems has been extensively discussed by seminal work of Cutkosky et al. [2023]. Unlike PGD Jin et al. [2017] which adds periodic perturbations sampled from a unit ball, MoXCo derives perturbations systematically through Eq.2. Furthermore, we incorporate momentum into each step, resulting in a distinctly different optimization trajectory. This design enables more effective optimization than PGD methods, as shown in Section 3.1. Additional discussion in Appendix A.1

**Adapting to local geometry** Although, efforts have been made to enhance exploration efficiency Malviya et al. [2023] Liu et al. [2023], there is a notable gap in addressing the need for stopping criteria. Lewkowycz et al. [2020] develops a connection between large learning rate & flatness of minima in SGD-trained models, and Xie et al. [2022] develops a method for escaping saddle points & flat minima selection using ADAM. Various works focus on momentum schedules Wang et al. [2022], but few recognize the need for a rapid exploration and committing phase like ours. O'donoghue and Candes [2015] for convex settings, whereas we operate in a non-convex, non-smooth context. Zhou et al. [2020a] focuses on proximal gradient parameter restarts for non-convex optimization, but our framework systematically indicates momentum hyper-parameter restarting based on local geometry. Additionally, Liu et al. [2023] uses Inertial Momentum in a federated learning setup for global convergence, while our generalized setting, employs it as one step in a two-step framework. Furthermore, Foret et al. [2020] (SAM) proposes simultaneous optimization of loss values and loss sharpness as a geometric approach to finding favorable local minima. However, our analysis demonstrates that these two metrics alone provide insufficient characterization of the loss landscape's local geometry. Detailed discussion in Appendix A.1.

## 2. Notation and Setup

We use standard notation where bold letters denote vectors and italic alphabets denote sets. We are interested in optimizing any objective function $f \colon \mathbb{R}^d \longrightarrow \mathbb{R}$. We make no assumptions about the differentiability of $f$, noting that it is non-convex and possibly non-smooth. Unlike the standard optimization goal of finding $\arg\min_x f(x)$, we seek a vaguely defined goal of **finding a sufficiently "flat" local minimum with sufficiently low objective value**. This formulation better suits machine

learning tasks where the optimization objective $f$ serves as a surrogate for the (often unknown) true objective $F$, as excessive optimization of $f$ may lead to worse solution on $F$, aka over-fitting. We will sometimes also consider composite optimization problem of minimizing $f + g$ for some regularizer $g$, for which a proximity operator is efficiently computable. Similar to before, the goal is not to minimize $f + g$ but rather to find a "good" solution that generalizes.

**Required auxiliary information.** Besides the typical first-order oracle that provides stochastic or noisy estimates of $f$ and its (sub)gradients, we are also given $f_{\text{target}}$ and $\lambda_{\text{target}}$, which defines what target objective value and flatness of interest. These two quantities can be used before, after, and during the optimization.

**Examples with such auxiliary information** We have two primary problems of interest that we will use to evaluate our algorithm.

**Training neural networks for regressions tasks.** In this problem, we consider ReLU networks $f_\theta$ is parameterized by $\theta$ with $L$ layers and $m$ neurons per layer. For $l = 0, 1, \ldots L - 1$, each layer defines an affine transformation $M^l$ with weights & biases such that $h_\theta(\mathbf{x}) = M^{(L)}(\mathbf{x}; \theta)$. The network is parameterized by $\theta = [W^{(1)}, b^{(1)}, W^{(2)}, b^{(2)}, ..., W^{(L)}, b^{(L)}]$ and associated with square loss so $f_\theta(\mathbf{x}) = \sum_{i=1}^{n}(h(x_i; \theta) - \hat{y}_i)^2$. We abuse the notation and use $f_\theta$ and $f_\theta(\mathbf{x})$ interchangeably.

Let the dataset be generated iid with $x_i \sim P$ and $y_i = \mathbb{E}[y|x_i] + \mathcal{N}(0, \sigma^2)$. The optimal target is to fit $\mathbb{E}[y|x_i]$, yielding an expected loss of $\sigma^2$. In overparameterized networks, the sharpness of minima crucially determines the right fit, with the Hessian's largest eigenvalue controlling the network's regularity in function space [Mulayoff et al., 2021, Nacson et al., 2022]. This relationship allows us to calibrate our optimization targets: we set $f_{\text{target}}$ to match the label noise level $\sigma^2$, and $\lambda_{\text{target}}$ to achieve the desired functional regularity of $\mathbb{E}[y|x]$.

**Quantizing neural networks.** The second case study is model quantization - converting neural networks to use quantized (e.g., binary) weights for faster inference. Starting from a full-precision pretrained model, we fine-tune to obtain a quantized version with comparable performance [Bai et al., 2019, Courbariaux et al., 2015]. Here, $f_{\text{target}}$ and $\lambda_{\text{target}}$ are naturally defined as the training loss and maximum Hessian eigenvalue of the initial full-precision weights.

# 3. MoXCo: Designing adaptive optimizers

We now describe the MoXCo algorithm outlined in Algorithm 1. As previously noted, the algorithm operates in two distinct phases (Phase 1 & Phase 2). Phase 1 focuses on exploration while monitoring certain statistics during optimization. The algorithm enters Phase 2 when it decides that the current parameter is in a good "ballpark" to commit to. Once in Phase 2, a different optimization algorithm is used to converge to the closest local minima as quickly as possible.

There are plenty of off-the-shelf optimization algorithm that solves Phase 2 and we recommend Hypergradient [Baydin et al., 2018] as a nearly hyperparameter-free method that works well. The remainder of the section focuses on two novel components in our MoXCo design: (1) the Inertial Proximal Algorithm for Promoting Exploration in Phase I in Section 3.1 and (2) the "Goodness" score we propose to adaptively determine the right time enter Phase 2 in Section 3.2.

## 3.1. Inertial Proximal Algorithm for Promoting Exploration

How do we promote exploration in deep learning training? One of the oldest idea is leveraging the inertial force of the "heavy-ball" algorithm [Polyak, 1964], known as "SGD with momentum" in deep learning optimization. Empirically, it has been observed that momentum accelerates SGD in low-curvature regions (e.g., plateaus) [Sutskever et al., 2013] and aids in escaping saddle points and shallow local minima [Wang et al., 2021].

However, vanilla SGD with momentum falls short in three aspects. First, its effectiveness on non-differentiable objective functions, is unclear — which is the case for every ReLU-activated neural networks. Second, it does not support the use of a proximal gradient-type method to handle structural

inducing regularization. Third, it does not leverage other tricks in deep learning optimizations, e.g., Adam, which are often delicate choices that enable the effective training of certain families of neural architecture.

The second problem was solved in the seminal work of Ochs et al. [2014] which establishes strong convergence guarantees for proximal versions of heavy-ball algorithm. To address the first problem, we proposed adding second momentum that slightly perturbs the location to evaluate the gradient as follows.

$$u_t = \theta_t + \alpha_t(\theta_t - \theta_{t-1}) \tag{1}$$
$$v_t = \theta_t + \beta_t(\theta_t - \theta_{t-1}) \tag{2}$$
$$\theta_{t+1} = \text{Prox}_g\left(u_t - \eta\hat{\nabla}f(v_t)\right) \tag{3}$$

where $\eta > 0$ is the learning rate and $\alpha, \beta \in [0, 1)$ are momentum coefficients and can vary with $t$.

The second line changes the location to evaluate the gradient slightly from $\theta_t$ to $v_t$. This is related to the recently proposed online-to-non-convex conversion method [Cutkosky et al., 2023, Remark 10] but without the randomized smoothing. The extra momentum on $v_t$ is particularly important because the proximal operator is very likely to return the subsequent iterate $\theta_{t+1}$ on highly-special non-smooth points, even if these non-differentiable points are inside a measure zero set. To address the third problem, we propose to think outside the box and apply the above algorithm to any deep learning optimizer via an interactive black-box fashion. In particular, we can return a different update for the iterates above by replacing $\hat{\nabla}f(v_t)$ with the update $\Delta_t$ sent back to us by any optimizer, i.e.,

$$\theta_{t+1} = \text{Prox}_g\left(u_t - \eta\Delta_t\right) \tag{4}$$
$$\Delta_t : \text{Optimizer}(\nabla f(v_t), v_t, \gamma) \tag{5}$$

where $\gamma$ represents any other inputs required for black-box optimizer, apart from $\nabla f(v_t)$. Also, the optimizer is allowed to have a memory from the previously observed $\nabla f(v_i), \boldsymbol{\theta}_i$ for $i \in [t-1]$.

Readers familiar with the Adam optimizer may ask "Wait a minute! Isn't momentum incorporated in Adam already? Why do I need a second round of momentum nested above in the outer loop?"

We argue that while both are termed momentum, they serve different purposes. Adam's momentum estimates moments for coordinate-wise scaling via exponential smoothing, adapting to local geometry. In contrast, our additional momentum promotes exploration independently of the black-box optimizer, preserving its hyper-parameters. The main advantage of doing so is that we can inherit the same hyper-parameter choices for the base optimizer black-box. Informally, $\Delta_t$ represents gradients of unknown functions induced by the tricks applied inside each black-box optimizer to regularize the energy landscape. Whereas, our proximal double inertial algorithm's parameters $\alpha, \beta$ control exploration-exploitation trade-off externally (low values signal interest to settle down and converge).

## 3.2. "Goodness" score and when to stop exploration

Having established inertial momentum for accelerated exploration, we address the second question: how to determine if the current local neighborhood merits our algorithm's commitment to convergence. This is a daunting task — using only local information (gradients and function values) we want to estimate a global property — namely, whether the current neighborhood contains a solution to the population-level stochastic optimization problem. While this is generally impossible for non-convex problems, deep learning offers unique structural properties and additional information (e.g., $f_{\text{target}}$, boundedness). We leverage these characteristics to formulate necessary conditions for "goodness" of a local neighborhood and use that as a promising heuristic to guide our optimizer.

Specifically, we came up with the following "goodness" score of a parameter $\theta$

$$\text{Goodness}(\theta) = \exp\left(-\tau\left[||\nabla f_\theta||_2^2 + \left|\frac{\lambda_{\max}(\nabla^2 f_\theta)}{\lambda_{\text{target}}} - 1\right| + |f_\theta - f_{\text{target}}|\right]\right) \tag{6}$$

in which $\tau$ calibrates how sensitive the score is, and the exponential transformation ensures that the score $0 \le \text{Goodness} \le 1$ as the term in the square brackets is nonzero. A goodness score being

Table 1: Optimization Landscapes. $f$ denotes any objective function parameterized by $\theta$. We enumerate magnitude of objective value, its hessian & largest eigenvalue at any $\theta$. For more refer A.2

| Indicator | Plateau | Cliff | Sharp | Saddle point |
|---|---|---|---|---|
| $f_\theta$ | large | large/small | small | large |
| $\|\nabla f_\theta\|_2^2$ | small | large | small | small |
| $\lambda_{\max}(\nabla^2 f_\theta)$ | small | small | large | small |

closer to one means that we have found a solution that is approximately stationary and close to an ideal target in both objective function value and flatness. (extended discussion: A.2, A.2.1)

**Theorem 1.** *Let $\theta$ obey that (a) it is a local minimum of $f$; (b) it is a stable fixed point of gradient descent with learning rate $\tilde{\eta}$, (c) $|f(\theta) - f_{target}| \leq \epsilon$. Then **Goodness**$(\theta) \geq \exp(-\tau\epsilon)$.*

**Theorem 2** (EOS adjustment)**.** *If we consider running Inertial momentum to optimize a quadratic objective function of form $f(x) = \frac{1}{2}\mathbf{x}^T\mathbf{A}\mathbf{x} + \mathbf{b}^T\mathbf{x} + c$, then based on Cohen et al. [2021], if we consider running vanilla gradient descent on the $f(x)$ starting from any initialization and if $(\mathbf{q}, a)$ be an eigenvector/eigenvalue pair of $\mathbf{A}$, then if $a > \frac{2(1+\alpha)}{\eta(1+2\beta)}$, then the sequence $\{\mathbf{q}^T\mathbf{x}_t\}$ will diverge.*

A desired learning algorithm should have $\epsilon \leftarrow 0$ as the number of data points $n$ gets larger, which means the Goodness score should converge to 1. (proof of Theorem 2 in A.3)

**Our algorithm** in conjunction with previous section, continuously monitors this score and reduces $\alpha, \beta$ once the score ($\delta$) exceeds a threshold $C$. The first-term ($\|\nabla f_\theta\|_2^2$) is the standard measurement of stationarity. The second-term $\left(\left|\frac{\lambda_{\max}(\nabla^2 f_\theta)}{\lambda_{\text{target}}} - 1\right|\right)$ measures $\theta$'s sharpness, normalized by $\lambda_{\text{target}}$ which depends on problem-specific curvature and should ideally match or exceed the designated "Edge of Stability" level of sharpness specified by the adjusted effective learning rate $\tilde{\eta}$ (Theorem 2). The third term ($|f_\theta - f_{\text{target}}|$) quantifies the absolute difference between the current objective and a problem-dependent target. For example, $f_{\text{target}}$ should be 0 for classification tasks, and $\sigma^2$ for regression.

Table 1 demonstrates the necessity of thresholding this score function for finding a "good solution". It demonstrates why a Goodness score metric is necessary - Given the diverse geometric properties of local minima in deep neural networks (plateaus, sharp valleys, etc.) Pascanu et al. [2014] Frye et al. [2020], no single metric can fully characterize a desirable minimum. We therefore analyze multiple geometric indicators: the loss value , gradient norm square, and the largest eigenvalue. As shown in Table 1, no one indicator always indicates a consistent value (small or large) across these pathological curvatures but by combining these metrics additively, we can always detect any non-desirable critical points. More details in A.2.

Building on the observed correlation between lower curvature and generalization performance, we conjecture that optimal performance occurs for a fixed computational budget. We find that this conjecture holds across all cases we tried, even when comparing different learning rates trained for the same duration. For empirical validation of the usefulness of this criterion, please refer Appendix A.5.1. Note that when $f$ is the training objective for a large dataset, monitoring the goodness score at every iteration is costly. We propose computing it periodically, such as at the end of each epoch, or incrementally estimating the statistics. More details are provided in A.2.2.

## 3.3. Dynamics of stopping and effect of Random Beta

In Phase 2, we reduce momentum to negligible values like 0.01 or 0.1 and adjust the step size adaptively using the Hyper-gradient descent method Baydin et al. [2018], which emulates physical deceleration towards a local minimum. In our analysis, inspired by Cutkosky et al. [2023], we examined the impact of random beta initialization for the second trajectory using toy examples and ResNets (see Section 4). Our results align with them, indicating that only the initial trajectory's momentum requires consideration.

---

**Algorithm 1:** MoXCo using Black-box Optimizer

---

**Input** : Max number of iterations $T_1, T_2$, Checkpoint frequency $N$, step size $\eta$, threshold $C$, objective function $f$, stochastic gradient oracle $\hat{\nabla} f$, initial parameter vector $\theta_0$, inertial momentum schedule $\alpha_t, \beta_t$ for $t = 0, 1, 2, ...$, Phase 1 optimizer: OPTIMIZER1, Phase 2 optimizer: OPTIMIZER2, Auxiliary $\lambda_{\text{target}}, f_{\text{target}}$.

**1** **Initialization** $\alpha_0, \beta_0, \theta_0, \theta_{-1} \longleftarrow \theta_0$

**Output:** Resulting parameters $\theta_t$

**2** **for** $t = 1, 2 \ldots T_1$ **do**

**3**     $u_{t-1} = \theta_{t-1} + \alpha_{t-1}(\theta_{t-1} - \theta_{t-2})$          // Phase 1 Exploration

**4**     $v_{t-1} = \theta_{t-1} + \beta_{t-1}(\theta_{t-1} - \theta_{t-2})$

**5**     $\Delta_t, \gamma_t = \text{OPTIMIZER1}(v_{t-1}, \nabla f(v_{t-1}), \gamma_{t-1})$

**6**     $\theta_t \longleftarrow u_{t-1,i} - \eta \Delta_t$          // Update parameters

**7**     **if** $t \bmod N = 0$ **then**

**8**        $\delta_t \longleftarrow \mathsf{Goodness}(\theta_t)$ as in (6) or any efficient approximation.    // goodness score

**9**        Break **if** $\delta_t > C$.

**10**     **end if**

**11** **end for**

**12** Reinitialize $\theta_0 \longleftarrow \theta_t$          // StartPhase 2

**13** **for** $t = 1, 2 \ldots T_2$ **do**

**14**     $\theta_t = \text{OPTIMIZER2}(\theta_{t-1}, \hat{\nabla} f(\theta_{t-1}))$

**15** **end for**

---

# 4. Experiments

In this section, we detail our experiments aimed to understand the behavior of MoXCo across different loss landscapes. We validate the following four claims (in bold) in subsequent paragraphs and describe the corresponding experimental setup and results. Any additional experimental details and ablations can be found in Appendix A.4, A.5.3.

**4(i)** **Aggressive exploration is helpful** Phase 1 helps in jumping over local suboptimalities when loss landscapes are plagued with several local minima.

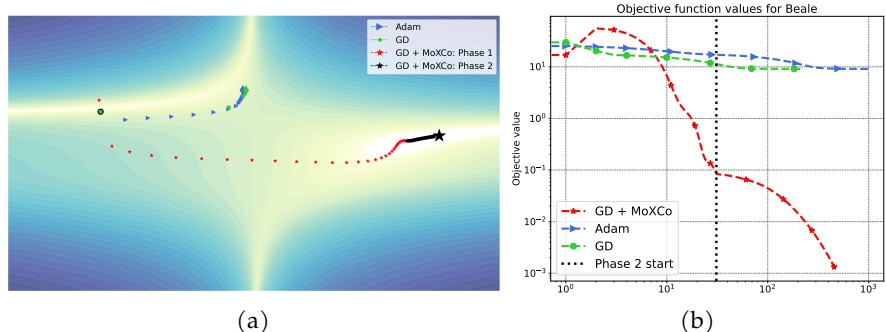

|     |     |
| --- | --- |
| (a) | (b) |

Figure 2: Trajectory visualization for the Beale: (a) MoXCo (red), with Phase 2 (black), successfully emerges from a poor initialization (far right) and reaches the global minimum marked $*$. Other two optimizers get stuck right after they start exploring. (b) Objective values vs.iterations (x-axis). MoXCo, with a smaller learning rate, converges as quickly as GD (faster than ADAM), despite both using $\eta = 0.01$. ADAM converges slower due to residual oscillations. Even with the same $\eta$, MoXCo converges faster A.5.3. Note: All runs till convergence.

**Setup:** We examine the impact of Phase 1 on the test optimization functions Beale and Ackley (A.4.1). We use MoXCo with GD and ADAM iterates, comparing trajectories against vanilla GD & ADAM. Highest possible stable learning rates used for all optimizers. Additionally, we plot objective values for all 3 optimizers to access their performance.

**Results** Figure 2(a) compares optimization trajectories of MoXCo-GD, Adam, and GD over 1000 steps on the Beale function, which contains three sub-optimal and one global minima. MoXCo, when initialized in high-energy regions, demonstrates two distinct phases: aggressive exploration

(red trajectory) followed by transition to (Phase 2) - convergence, i.e reaching a flatter region—as determined by the goodness score. This two-phase approach enables MoXCo to escape sub-optimal minima and achieve better convergence than competitors that become trapped locally.

Figures 1 and 2 demonstrate similar behavior on Ackley and Beale functions over 1000 steps. Fig1 uses consistent and highest stable learning rate of $\eta = 0.05$, with $\alpha = 0.99$. For Fig 2, $\eta = 0.005$ with $\alpha = 0.8/0.9$ ($\beta$ is random for all). Vanilla optimizers use $\eta = 0.01/0.05$. MoXCo (red curve) maintains effective exploration-convergence balance despite significant momentum and a relatively large $\eta$. Figure 2(b) shows MoXCo achieves comparable or better convergence speeds despite using lower learning rates than GD and ADAM, with superior performance (faster speed of convergence & better quality of solutions) when using identical learning rates (Fig 9) [additional experiments including MoXCo+ADAM in A.5.3, details in A.4.1].

**4(ii)  "Goodness score" accurately gauges optimal regions in a solution space** Phase 2 helps in effectively evading sharp basins and converge to flat loss region by adapting to the local geometry and we validate this as described below.

**Setup** To understand more on behavior of goodness score, we plot final MSE values attained when MoXCo Phase 2 begins at different thresholds $\gamma$. (see A.2.1, A.2, A.2.3 for more details)

**Results** This claim in partly supported by Section 4(i) and Section 4(iii) results. Additional experiments in Appendix A.5.3. For the results of the setup discussed above, refer Appendix A.5.1

**4(iii)  Both phases improve outcomes across complex loss landscapes** We can empirically validate that MoXCo framework leads to faster convergence speeds and reaches flatter local minima.

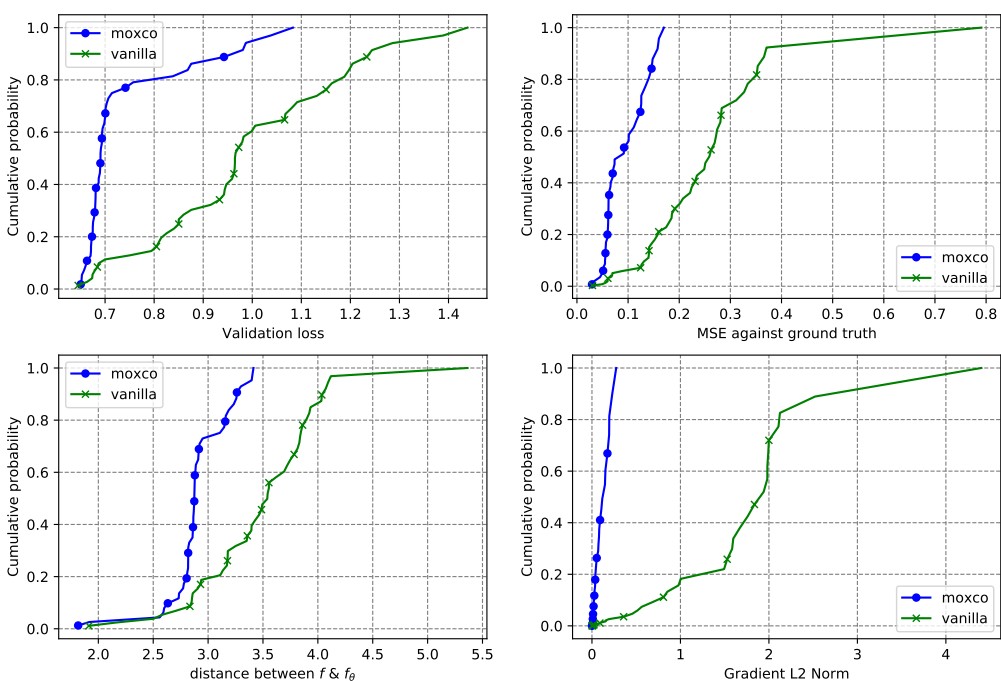

Figure 3: Hit rate analysis for 50 random initialization when $m = 5$ for under-parameterized two-layer NN optimized using MoXCo + GD. Dataset consists of 30 data points with noisy labels (see A.5.4). We evaluate "goodness" of solutions at convergence for these 50 runs using these 4 metrics. All four metrics are significantly smaller for MoXCo as compared to vanilla GD.

**Setup** We analyze a two-layer ReLU network with quadratic loss under both under- and over-parameterized settings with black-box optimizers. Our analysis uses 30 data points where $x_j \sim \mathcal{U}[0, 1]$ and observations $\hat{y}_i = f(x_i) + \epsilon_i$ with $\epsilon_i \sim \mathcal{N}(0, 0.8^2)$ (details in 2, A.5.4). Despite being a shallow network, ReLU activations introduce non-smoothness and non-convexity. For each setting, we

conduct a hit-rate analysis by analyzing performance via Cumulative distribution function (CDF) plots across 50 initializations with varying neuron counts $m$ (A.4.3), comparing fitted functions against target $f$ (details in A.4.2).

*Under-parameterized case:* The function $f$ that we try to estimate is a simple piece-wise function as in Fig 4 (A.4.2). This setting is ideal for testing our claims, as under-parameterized loss landscapes are inherently complex. They often feature multiple isolated local minima with positive definite Hessians, reflecting local convexity Liu et al. [2021]. In contrast, over-parameterized settings typically lack convexity in any neighborhood around a global minimizer. We report results with $m = 5$ in Fig 3, 4 & m=2 in A.5.4. Consistent setting with fixed $\alpha, \beta = 0.6, 0.6, \eta = 0.1$ are used throughout the results in under-parameterized regime.

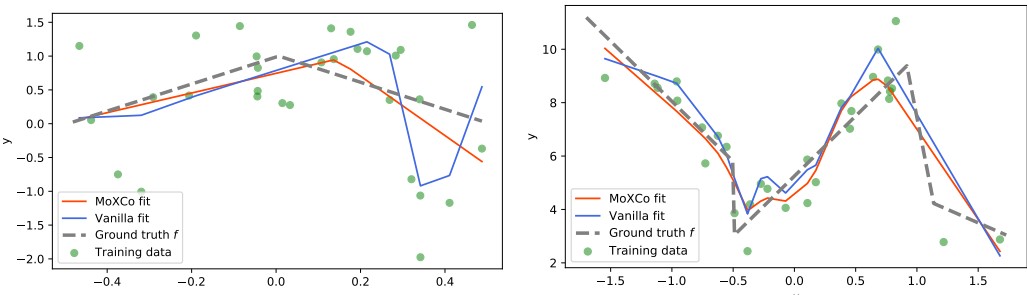

Figure 4: Learned functions for over-parameterized and under-parameterized regimes. (a) An example run among random 50 initializations of Figure 4 ($m = 5$). For the same $\eta$, GD fits to a more complex solution and learns noisy labels as compared to MoXCo fit. Thus, demonstrating that MoXCo learned a solution that is better than GD.(b) Similar behavior observed for over-parameterized network case ($m = 20000$). See A.4.2 for details.

*Over-parameterized case* We use a different function $f$ as described in A.4.2. Due to the essential non-convexity of loss landscapes in overparameterized setups, this setting is not ideal for understanding the two phases of our algorithm. Therefore, our analysis primarily focuses on the under-parameterized setup. Here, $\alpha, \beta = 0.9, 0.9, \eta = 0.05$.

**Metrics** We evaluate performance using three metrics: (1) MSE against validation and ground truth for generalization error, (2) gradient $L^2$ norm to assess convergence stability, and (3) $L^2$ distance between the learned function $f_\theta$ and ground truth $f$. Since both $f$ and $f_\theta$ are bounded and defined on $[-0.5, 0.5]$, they are Riemann integrable, allowing us to compute their $L^2$ difference through Riemann sum approximations (see A.4). Smaller values in all metrics indicate better performance: lower generalization error, stable local minima, and better function approximation respectively.

**Results** Figure 3 demonstrates MoXCo's superior performance over vanilla GD in underparameterized settings ($n > m$), consistently achieving lower values across all four metrics. MoXCo converges to flatter minima regardless of initialization, and as shown in Fig 4, learns better approximations to $f$ while avoiding noise fitting in both under- and over-parameterized regimes ($m < n$ and $m > n$) [additional experiments in A.5.4, details in A.4.2].

**4(iv) Application on ResNets** MoXCo is applicable to general deep learning tasks and we find that our method is particularly effective in scenarios with highly irregular loss geometries. That is, while standard ResNet training involves relatively smoother loss surfaces, quantization introduces significant non-smoothness and discrete constraints, resulting in highly non-convex landscapes with numerous local minima which makes learning in ResNet a lot harder. Under this setting, below we demonstrate MoXCo's efficacy in achieving better solutions & enhancing generalization performance compared to vanilla PROXQUANT method.

**Setup** We apply MoXCo to binary quantization of residual networks using PROXQUANT Bai et al. [2019]. Despite non-convexity and non-smoothness due the quantization-inducing regularizer, we integrate MoXCo using equations (4,5) with the proximal operator. We compare against PROX-

QUANT with SGD (base optimizer), using tuned hyper-parameters for both methods and the base floating-point (FP) model. Our implementation uses a homotopy scheme ($\lambda = 10^{-5}t$) without learning rate scheduling (details in A.4.5).

**Results** MoXCo integrated with PROXQUANT achieves improved Top-1 classification precision across all depth compared to using vanilla PROXQUANT.

Table 2: Top-1 classification precision of binarized ResNets on CIFAR-10. Performance is reported in mean(std) over 4 runs, as well as the (absolute) performance boost of over PROXQUANT.

| | | Classification precision | | Performance boost over PQ-B net |
|---|---|---|---|---|
| Model (Bits) | FP (32) | PQ-B (1) | MoXCo-PQ (ours) (1) | MoXCo-PQ (ours) (1) |
| ResNet-20 | 87.63 | 77.11 (0.07) | 80.34 (0.15) | **+3.23** |
| ResNet-32 | 88.17 | 79.91 (0.03) | 81.79 (0.06) | **+1.88** |
| ResNet-44 | 88.98 | 80.55 (0.04) | 82.05 (0.08) | **+1.55** |
| ResNet-56 | 88.55 | 80.53 (0.06) | 82.47 (0.05) | **+1.94** |

## 5. Limitations and Future Work

We acknowledge limitations and several key directions for future research that would strengthen our work. First, while we propose a computationally efficient variant of our method in Section A.2.1, comprehensive empirical validation remains to be conducted. Second, a systematic comparison between our approach and other aggressive exploration methods, particularly those using very large learning rates, would provide valuable insights into the relative merits of each strategy. Finally, extensive large-scale experiments would help establish stronger benchmarks and better demonstrate the scalability of our method across diverse optimization scenarios.

## 6. Conclusion

In this paper, we propose and evaluate MoXCo, an adaptive method for optimizing complex loss landscapes. Our results demonstrate that MoXCo provides a novel perspective on optimization theory by focusing not solely on minimizing loss functions but on identifying favorable solutions within any loss landscape. For future work, it would be valuable to explore and enhance the computational aspects of monitoring the "goodness" score for real-time deep learning applications and to investigate the effects of learning rate schedules.

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

# A. Appendix

## Table of Contents

## A.1. Extended Related Work

**PGD vs MoXCo:** Our two-momentum approach uniquely combines acceleration and exploration: primary momentum (Eq. 1) smooths the optimization trajectory, while secondary momentum (Eq. 2) computes gradients at perturbed points.

**SAM vs MoXCo**: We argue for incorporating additional geometric indicators beyond sharpness and loss magnitude to reliably distinguish between local minima quality, as evidenced by our empirical results in Table 1. Additionally, sharpness at each optimization step is computationally expensive. Instead, we propose using sharpness as one of other heuristic indicators to characterize reliably local geometry and in extension a local minima. Informally, our approach is more "locally" adaptive and efficient since we approximate curvature early in optimization, then efficiently converges to flat local minima through increased step sizes Qiao et al. [2024].

## A.2. Phase 2: More on Heuristic function

Our method exploits the local geometric properties of the objective function such as sharpness, strong convexity, & heterogeneous curvature in different directions to effectively escape sharp basins and converge to flat loss regions Zhou et al. [2020b]. We design measures of such relevant information and refer to them as "indicators". The biggest benefit of our methods is that they incur no additional computational cost. This is because all the necessary local information either already exists or can be readily computed at any given time step.

**Measure of "flatness"** There is no universally accepted definition for flat minima, and the concept may carry slightly varying interpretations across different works Dinh et al. [2017] Hochreiter and Schmidhuber [1997]. We define sharpness via the eigenspectrum of the Hessian of the training objective, particularly considering the largest eigenvalue $\lambda_{max}(\nabla^2 f(\boldsymbol{\theta}_t))$, a commonly adopted metric Jastrzebski et al. [2017] Dinh et al. [2017] Kaur et al. [2023]. Since it is established that flatness is linked to better generalization Keskar et al. [2016], it is crucial to analyze the geometric properties of the loss landscape. The mean curvature in the original loss space determines whether saddle points, on average, appear as minima, maxima, or almost flat regions Böttcher and Wheeler [2022]. To efficiently estimate the local curvature information at any given time step during training, we employ the HessianFlow method. This technique computes the top eigenvalues of the Hessian operator using a matrix-free algorithm, as outlined in Yao et al. [2018a]Yao et al. [2018b]. In our methodology, the Hessian spectrum is evaluated only at the conclusion of each epoch, resulting in minimal computational overhead. A smaller eigenvalue serves as an indicator of proximity to a flatter region. Table 1 illustrates the potential range of eigenvalues corresponding to various irregularities present in a loss landscape.

**Local Geometric Curvature properties:** Examining geometric attributes of the loss landscape, including local curvature and the presence of alternative optima in the vicinity of a specific point in loss space, is often associated with enhancing the performance and generalization capabilities of neural networks Böttcher and Wheeler [2022]Keskar et al. [2016]Hochreiter and Schmidhuber [1997]Wu et al. [2018]. Therefore, to obtain a precise assessment of pathological curvatures in the loss landscapes, we consider the l2 gradient norm of the model parameters. Low gradient information indicates the proximity of potential critical points. When combined with the largest eigenvalue, we can make reasonable inferences about proximal irregularities with a relatively high degree of confidence. Also, it's important to note that estimating the l2 gradient norm can be challenging due to the inherent noise in the observed gradients during training. This noise can potentially skew the results, which is why we consider a denoising (and de-biasing) approach for our gradient l2 norm estimates. We use universal dynamic regret to exploit local strong convexity by using ALIGATOR Baby et al. [2021] to find a true estimate of gradient l2 norm.

**Measure of Convergence** A low objective function value (zero or near-zero training loss) can signify either proximity to the optimal solution or a local minimum. In non-convex problems, such as the one we're addressing, this ambiguity is even more pronounced because our objective is to attain an optimal solution that exhibits better generalization. Being situated near a sub-optimal minimum is undesirable. This also justifies the necessity of employing multiple indicators or 'sensors' of the loss landscape, as a low objective function value can be highly misleading. In our experiments [section 5], with binary quantization, we have a non-smooth regularized objective function, which serves as one of our indicator measures.

**Restart window determination** Recommended range for $\delta$ lies in range [0.80 to 0.95]. Increasing values of $\delta$ signify proximity to flatter minima. As $\delta$ increases, we are closer to converging to a stationary point, and by virtue of our heuristic function, with high probability, these stationary points will be situated in desired flatter regions. Thereby indicating high confidence for momentum reset. Similar to the findings of Hinton et al. [2006], we observed that resetting momentum parameters towards the end of convergence is advantageous. In terms of a physics analogy, the heuristic function's estimate of $\delta$ helps determine when we should begin decelerating as we approach a shallow minima.

**Table 1 analysis:** Table 1 demonstrates why a Goodness score metric is necessary - Given the diverse geometric properties of local minima in deep neural networks (plateaus, sharp valleys, etc.) Pascanu et al. [2014] Frye et al. [2020], no single metric can fully characterize a desirable minimum. We therefore analyze multiple geometric indicators: the loss value $f_\theta$, gradient norm square $||\nabla f_\theta||_2^2$, and the largest eigenvalue $\lambda_{\max}(\nabla^2 f_\theta)$. As shown in Table 1, no one indicator always indicates a consistent value (small or large) across these pathological curvatures but by combining these metrics additively, we can always detect any of non-desirable critical points.

The desirable local minima is the one which generalizes - a wide & flat local minima, which we found can be categorized by all three indicators having a small value - as it reaches a stable stationary

point $f_\theta$ and $\|\nabla f_\theta\|_2^2$ (small) whereas wide and flatness implies a small $\lambda_{\max}(\nabla^2 f_\theta)$. By combining these three metrics, we can reliably characterize geometry of local minimas' that offers strong generalization properties. When all three indicators show small values, we consistently find good minima, independent of the loss landscape's structure.

### A.2.1. Eigenvalue Estimation

To estimate the maximum eigenvalue of the hessian operator at every time step efficiently, we use the method implemented by Yao et al. [2020, 2018a]. This is a power iteration-style method that iteratively computes Hessian-vector product and re-normalizing to find the eigenvector $v$ and eigenvalue $\lambda$ that satisfies $\lambda v = Hv$. The Hessian vector product (HVP) can be computed using backpropagation without explicitly constructing the Hessian matrix. Dagréou et al. [2024] provided a concise review of HVP and its computational and memory overhead in modern deep learning models.

Below in Algorithm 2, we describe how the sharpness measure we need for the Goodness score is computed. Throughout all our experiments, we assume $\lambda_{\text{target}} \geq k*(\text{EOS}_{\text{adjusted}})$, where $k \geq 1$.

---

**Algorithm 2:** MoXCo local measure of sharpness for parameter $\theta$

---
1 **Initialization** *random vector $v$*
  **Input** :$\lambda_{\text{target}}$, Iterations $T$, Stochastic objective $f$, parameter $\theta$
  **Output**:Normalized local measures of curvature $\nu$ at current epoch
2 **for** $i = 1, 2, \ldots, T$ **do**
3     $Hv \leftarrow \nabla^2 f(\theta)v$ (using efficient Hv product)
4     $v \leftarrow Hv/\|Hv\|_2$
5 **end for**
6 $\lambda_{\max} \leftarrow v^T \nabla^2 f(\theta)v$ (using efficient Hv product)
7 Return $\nu = \frac{|\lambda_{\max} - \lambda_{\text{target}}|}{\lambda_{\text{target}}}$

---

We also propose two tricks to make the power iterations above more efficient.

First, replace the full Hessian Vector product $\nabla^2 f(\theta)v$ with a stochastic approximation version of the Hessian Vector product $\nabla^2 \hat{f}(\theta)v$ where $\nabla^2 \hat{f}$ is computed from a randomly chosen minibatch. This still correctly computes the largest eigenvalue as the number of iterations gets large Hardt and Price [2014].

Second, we propose to run only one iteration of the power iteration each time and use the previous iteration's output eigenvector as an initialization, namely, warm start. Even though $\nabla^2 f(\theta)$ changes with $\theta$, this algorithm is already tracking the exact computation and it significantly lowers the overhead of computing the "Goodness" score.

### A.2.2. Function value and Gradient Square Estimation

Besides $\lambda_{\max}$, we also need the function value $f(\theta)$ and gradient norm $\|\nabla f(\theta)\|^2$ when evaluating the Goodness score.

When $f$ is a machine learning training objective function, both the function value and gradient norm require $O(n)$ in the number of data points $n$. This becomes too much overhead to compute in every iteration. One could, potentially compute them infrequently, i.e., only after each full data pass. This works when the optimization algorithm runs for many epochs, but is too infrequent when only a few data passes is scheduled.

We propose a minibatched stochastic approximation that provides more frequent approximate measurement of $f(\theta_t)$ and $\|\widehat{\nabla f(\theta_t)}\|^2$ for every iteration $t$.

**Estimating** $f(\theta_t)$**.** The idea is similar to the standard exponential weighted averages in online smoothing by outputting

$$\widehat{f(\theta_t)} = \frac{1}{1-\gamma} \sum_{i=t,t-1,\dots} \gamma^{t-i} \hat{f}_i(\theta_i)$$

for a choice of $\gamma$ close to 1, e.g. 0.99. $\hat{f}_i\theta_i$ is the average loss computed on the minibatch of iteration $i$.

This approach works but we need to choose $\gamma$ and there may not be a good constant choice for the effective "window size" $1/(1-\gamma)$ in the cases when we are running an aggressive exploration algorithm in Phase I of MoXCO. Some local region may need a smaller window size because the $f(\theta)$ changes too quickly. Other regions may prefer a bigger window size because $f(\theta)$ is relatively flat and bigger window size gives estimators with lower variance.

To solve this problem, we propose using a parameter-free adaptive online smoothing algorithm known as ALIGATOR [Baby et al., 2021]. ALIGATOR takes an arbitrary sequence of noisy observations $y_i = \mu_i + \text{noise}_i$ for $i = 1, 2, 3, \dots$ and provides a nearly optimal online estimate to $\mu_i$ without knowing how quickly $\mu_i$ changes over $i$. It can essentially compete with an oracle choice of the best window size chosen at any location $i$, thus suitable for our need.

To be concrete, we pass $\hat{f}_i\theta_i$ toe ALIGATOR and ALIGATOR returns a good estimator of $\hat{f}(\theta_i)$ on the fly that we plug into the Goodness score.

**Estimating** $\|\nabla f(\theta_t)\|^2$**.** The issue is similar but slightly more complex for the gradient norm estimate. One naive approach is to use ALIGATOR to estimate every coordinate of the gradient separately and then plug in. But this will over-estimate the gradient norm because every coordinate's estimate is noisy. The over-estimate is on the order of $O(d\sigma^2)$ where $d$ is the dimensionality. This can be painful even if ALIGATOR is able to suppress $\sigma^2$ to $1/\text{Optimal\_Window\_Size}$.

Instead, we propose estimating the gradient norm square directly by estimating the RHS of the following decomposition

$$\|\nabla f(\theta_t)\|^2 = \|\nabla \hat{f}(\theta_t) - \nabla f(\theta_t)\|^2 + \|\nabla \hat{f}(\theta_t)\|^2 + 2(\nabla \hat{f}(\theta_t) - \nabla f(\theta_t))^T \nabla \hat{f}(\theta_t)$$

$\|\nabla \hat{f}(\theta_t) - \nabla f(\theta_t)\|^2$ is the variance that can be unbiasedly estimated since we have a minibatch (with minibatch size $> 1$). We can run ALIGATOR for this.

$\|\nabla \hat{f}(\theta_t)\|^2$ is directly observable, thus we can run ALIGATOR to denoise the this sequence.

As for the cross term, $2(\nabla \hat{f}(\theta_t) - \nabla f(\theta_t))^T \nabla \hat{f}(\theta_t)$, we can construct an unbiased estimator by using the symmetrization trick and data splitting. Basically, we can randomly split the minibatch into three parts, and the following is an unbiased estimator $2(\nabla \hat{f}_1(\theta_t) - \nabla \hat{f}_2(\theta_t))^T \nabla \hat{f}_3(\theta_t))$ Then we can smooth this sequence using ALIGATOR.

This ensures that there are just three ALIGATORs to run instead of $d$. and we do not suffer from $O(d)$ error in our estimate of the gradient norm square.

Note that, since we deal with GD in 4(i), 4(ii), we use a simple gradient $L^2$ norm estimate for both those set of experiments.

### A.2.3. Tuning hyper-parameters and estimating Goodness Score

MoXCo introduces two key hyper-parameters: the inertial momentum parameter $\alpha$ and the temperature $\tau$ for the goodness score criteria. $\tau$ and the threshold $\gamma$ for triggering the goodness score are inversely proportional, allowing for estimation of one from the other. Tuning $\alpha$ is akin to adjusting any momentum parameter, where a higher $\alpha$ promotes aggressive exploration and a lower $\alpha$, indicating values in the range $[0, 1)$, supports more conservative exploration. We recommend fine-tuning $\tau$ while keeping $\gamma$ within a fixed range of $[0.85, 0.95]$. Further insights into the effectiveness of the goodness score are illustrated in Figure 5, which demonstrates that the goodness score is activated

only when all three criteria are minimal(sharpness should be near EOS). Additionally, note that the squared norm of the gradient for the GD version of this experiment, shown in green, is diverging.

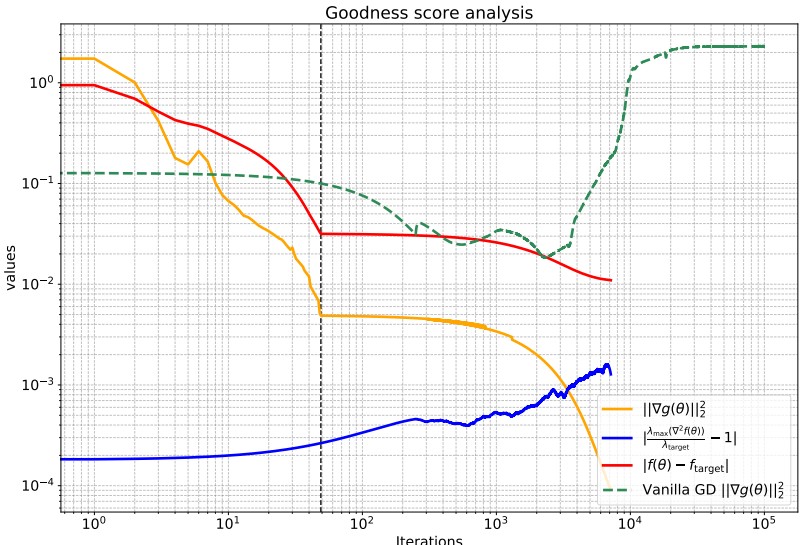

Figure 5: All three measures of goodness score plotted against iterations for initialization and setting similar to Figure 4(a)

## A.3. Stability properties of gradient descent (Inertial momentum variant)

Below are the steps used to derive the adjusted learning rate bound based on Edge of Stability Cohen et al. [2021] but for inertial momentum.

Optimizing against quadratic objective function :

$$f(x) = \frac{1}{2}\mathbf{x}^T \mathbf{A}\mathbf{x} + \mathbf{b}^T \mathbf{x} + c$$

Inertial equations:

$$\mathbf{v}_t = \mathbf{x}_t + \alpha(\mathbf{x}_t - \mathbf{x}_{t-1}) = (\alpha + 1)\mathbf{x}_t - \alpha\mathbf{x}_{t-1} \tag{7}$$

$$\mathbf{y}_t = x_t + \beta(\mathbf{x}_t - \mathbf{x}_{t-1}) \tag{8}$$

$$\mathbf{x}_{t+1} = \mathbf{v}_t - \eta\nabla f(\mathbf{y}_t) \tag{9}$$

$$\nabla_{y_t} f(\mathbf{y}_t) = \mathbf{A}((1 + \beta)\mathbf{x}_t - \beta\mathbf{x}_{t-1}) + \mathbf{b}$$
$$\nabla_{y_t} f(\mathbf{y}_t) = (1 + \beta)\mathbf{A}\mathbf{x}_t - \beta\mathbf{A}\mathbf{x}_{t-1} + \mathbf{b}$$

GD with inertial momentum update step wrt to the quadratic equation -

$$\mathbf{x}_{t+1} = \mathbf{v}_t - \eta\nabla L(\mathbf{y}_t)$$
$$\mathbf{x}_{t+1} = (\alpha + 1)\mathbf{x}_t - \alpha\mathbf{x}_{t-1} - \eta[(1 + \beta)\mathbf{A}\mathbf{x}_t - \beta\mathbf{A}\mathbf{x}_{t-1} + \mathbf{b}]$$
$$= [1 + \alpha - \eta(1 + \beta)\mathbf{A}]\mathbf{x}_t - (\alpha - \eta\beta\mathbf{A})\mathbf{x}_{t-1} + \mathbf{b}$$
$$= (1 + \alpha)\Big[\mathbf{I} - \frac{\eta(1 + \beta)}{1 + \alpha}\mathbf{A}\Big]\mathbf{x}_t - \alpha\Big[\mathbf{I} - \frac{\eta\beta\mathbf{A}}{\alpha}\Big]\mathbf{x}_{t-1} - \eta\mathbf{b}$$

The quantity $\mathbf{q}^T\mathbf{x}_t$ evolves under gradient descent as:- (also note $(\mathbf{q}^T\mathbf{A} = a\mathbf{q})$)

$$\mathbf{q}^T\mathbf{x}_{t+1} = (1 + \alpha)\Big[1 - \frac{\eta(1 + \beta)}{1 + \alpha}a\Big]\mathbf{q}^T\mathbf{x}_t - \alpha\Big[1 - \frac{\eta\beta a}{\alpha}\Big]\mathbf{q}^T\mathbf{x}_{t-1} - \eta\mathbf{q}^T\mathbf{b}$$

If we define $\tilde{x}_t = q^T x_t + \frac{q^T b}{a}$, and note that $q^T x_t$ diverges iff $\tilde{x}_t$ diverges. Thus, rearranging above equation,

$$\tilde{\mathbf{x}}_{t+1} = (1+\alpha)\Big[1 - \frac{\eta(1+\beta)}{1+\alpha}a\Big]\mathbf{q}^T\tilde{\mathbf{x}}_t - \alpha\Big[1 - \frac{\eta\beta a}{\alpha}\Big]\mathbf{q}^T\tilde{\mathbf{x}}_{t-1}$$

Above equation, is a linear homogeneous second-order difference equation. By Theorem 2.37 in Elaydi [2005] -

$$p_1 = (1+\alpha)\Big[1 - \frac{\eta(1+\beta)}{1+\alpha}a\Big]$$
$$p_2 = \alpha\Big[1 - \frac{\eta\beta a}{\alpha}\Big]$$

If $a > \frac{1}{\eta}\left(\frac{2+2\alpha}{1+2\beta}\right)$ then this recurrence diverges.

## A.4. Experimental Details

We consistently used a threshold $C$ in the range of $[0.80, 0.88]$ across all experiments. Depending on $C$, we tuned the temperature hyper-parameter ($\tau$) or adjusted $C$ based on $\eta$. Specifically, we found that $\tau = 0.15$ or $\tau = 0.2$ reliably estimated a threshold score of $0.85$ throughout our experiments. We consistently used $\beta_{\text{hypergradient}} = 1 \times 10^{-3}$ in all experiments, except in Section 4, where we used $\beta_{\text{hypergradient}} = 1 \times 10^{-2}$. The results did not significantly deviate when using the former value.

### A.4.1. Test Optimization Functions

In this section, we include additional experimental details for our results in Section 4.

The functional form for Beale is defined as -

$$F(\boldsymbol{x}) = (1.5 - x_1 + x_1 x_2)^2 + (2.25 - x_1 + x_1 x_2^2)^2 + (2.625 - x_1 x_1 x_2^3)^2$$

Ackley loss landscape serves as a good synthetic benchmark because it closely approximates a complex loss landscape surrounded with numerous local minima. -

$$F(\boldsymbol{x}) = -a\exp\Big(-b\sqrt{\frac{1}{d}\sum_{i=1}^{d}x_i^2}\Big) - \exp\Big(\frac{1}{d}\sum_{i=1}^{d}cos(cx_i)\Big) + a + \exp(1)$$
$$a = 20, b = 0.2, c = 2\pi$$

We used a goodness score threshold of $C = 0.85, 0.80$ respectively for Figures **??**. We include additional results on Ackley and Beale in next section, where we used similar configuration of $\tau, C$.

### A.4.2. Non-parametric Regression

For the non-parametric set of experiment, the two-layer neural network is used to approximate any parametric function $f$, with data points $\{\mathbf{x}_i\}_{i=1}^n$ where $n$ is the number of data points, each $x_j$ for $j = 0, 1, ...d$ is drawn i.i.d from an uniform distribution over $[0, 1]$. The number of neurons is denoted by $m$, characterizes model's complexity. We observe $\hat{y}_i = f(\mathbf{x}_i) + \epsilon_i$ where $\epsilon_i$ represents Gaussian noise. We study MoXCo's behavior under both under-parameterized ($n > m$) and over-parameterized regimes Soudry and Carmon [2016]. We adhere to the Neural Tangent Kernel (NTK) parameterization Jacot et al. [2020].

Functional form $f$ used in Section 4(ii) for experimenting with MoXCo when $m < n$ -

$$\begin{cases} 2x & x < 0.5 \\ -2x + 2 & x \geq 0.5 \end{cases}$$

Functional form $f$ used in Section 4(iii) for experimenting with MoXCo when $m > n$ -

$$\begin{cases} 2.55 - 4.5x & -2 \le x < -0.5 \\ -0.75 + 4.5x & -0.5 \le x < 1 \\ 5.5 - 2x & 1 \le x \end{cases}$$

This functional form has been studied in Mammen and Van De Geer [1997] in the context of understanding the effects of spatially inhomogeneous smoothness on least squares penalized regression estimates with total variation penalties. We adopt one of their functionals to test whether MoXCo for a regression task, can effectively learn the underlying function in an over-parameterized setting.

For the goodness score threshold, we consistently use $C = 0.85$ across all non-parametric experiments. In under-parameterized experiments, we use fixed settings of $\alpha = 0.6$, $\beta = 0.6$, $\eta = 0.1$, and $\tau = 0.20$. For over-parameterized experiments, we use $\eta = 0.05$, $\alpha = 0.9$, and $\beta = 0.9$. We used consistent $\beta_{\text{hypergradient}} = 1e - 3$ throughout all experiments. Note that all these results are extensible to cases where $\beta$ is random.

### A.4.3. Cumulative Distribution Function of performance metrics

In plots of 4(iii)show the Cumulative Distribution Function (CDF) of performance metrics (MSE, gradient norm, etc.) across 50 random initializations. For any threshold T, the CDF value indicates the fraction of initializations achieving performance $\le$ T. For example, CDF(0.8) = 0.7 means 70% of initializations achieved performance $\le 0.8$ (equivalently, 30% achieved $> 0.8$), providing insight into the training process's reliability across different initializations. In these figures, having a higher CDF value at lower thresholds is preferable - notably, MoXCo achieves substantially higher CDFs at smaller thresholds compared to vanilla GD across all metrics, indicating more reliable performance over multiple random initializations. We will also add this explanation in the revision.

### A.4.4. Distance Metric between functions

To qualitatively assess the performance of a neural network in capturing the underlying function, we measure the discrepancy between the predicted and true functions in the functional space. This measure of distance is equivalent to the $L2$ norm of their difference, which can be derived from the inner product of their difference. By definition of the inner product for continuous spaces, we can use Riemann integrals if the functions are Riemann integrable. To estimate this empirically in a discrete space, we use Riemann sum estimation. We employ this metric in our experimental evaluation in Section 4(ii)

### A.4.5. Application on ResNets with PROXQUANT

We used PROXQUANT algorithm publicly provided by Bai et al. [2019]. All MoXCo-PROXQUANT experiments use the same configurations for a fair comparison.We use a momentum schedule for $\alpha_t = \frac{3t-1}{3t+1}$, where $t$ refers to each iteration. Similarly, $\beta$ is set randomly at every iteration. We use a fixed step size $\eta = 0.025$, with initial values of $\alpha_0 = 0.5$ and $\beta_0 = 0.9$. Additionally, we set $\delta = 0.87$, and $\tau = 0.15$ and restart momentum parameters with $\alpha = 0.7, \beta = 0.3$. The step size $\eta$ is adapted using hyper-gradient.

Additionally, learning rate schedules do not pair well with our approach, as we employ an adaptive momentum framework. This framework includes phase 2 that restarts all momentum parameters and readjusts the learning rate adaptively as well. With respect to computational resources a single GPU is more than enough to run are experiments since they we work with ResNets of maximum depth of 56. Using MoXCo does not add additional computation complexity to it.

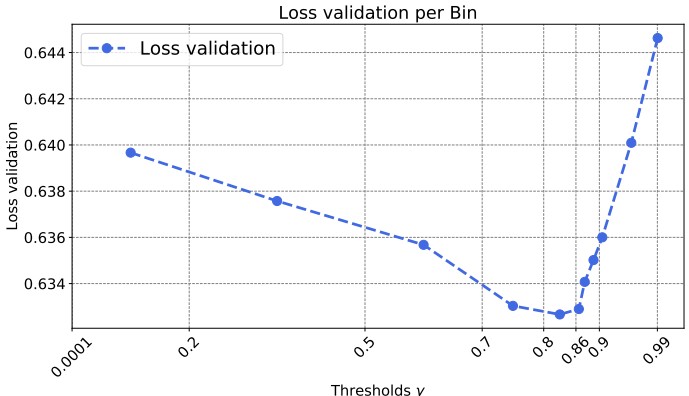

Figure 6: Goodness score v/s Loss validation to measure its effectiveness in identifying interesting local curvatures. Validation loss is smallest in the region of $[0.75 - 0.95)$. These experiments were done under over-parameterized setting of parametric function.

## A.5. Additional Experiments

### A.5.1. Results from Section 4(ii)

**Setup** To provide empirical validation of the usefulness of this goodness score criterion, we correlate it with population-level sub-optimality. We plot 50 goodness scores ranging from $[1e - 4, 1]$ and report the final loss validation achieved at convergence when MoXCo starts at the corresponding goodness scores.

**Results** Based on theoretical understanding and intuition about goodness scores, performance is expected to be poor at the extremes—when the goodness score is 0 and 1. It decreases from both sides centering or stabilizing near 0.80-0.95 range. At a goodness score of 1, Phase 2 is never active, meaning the algorithm relies solely on momentum exploration. We might overreach and pass by good localities and are a risk of divergence. Conversely, at a goodness score of 0, Phase 1 never starts, never initiates, eliminating the opportunity for aggressive exploration of the surrounding vicinity, which is also sub-optimal. In this case, the algorithm takes overly cautious steps and prepares for convergence prematurely. Hence, a U-shaped like curve is the expected behavior and we are able to see that empirically too. We used same setting as in under-parameterized analysis with $\alpha, \beta = 0.9, \tau = 0.2$. Only $\eta = 0.005$.

### A.5.2. Additional ablation for momentum parameter $\alpha$

Below, depicts ablation plot for our momentum parameter $\alpha$ and random $\beta$. This is in-line with our general recommendation of the $\beta$ which is to be randomly sampled between [0,1]. We ablate $\alpha = [0, 1)$, for under-parameterized (m=5) 2NN regression task. We report loss validation (MSE) averaged for five runs per $\alpha$. All the settings are exactly similar to experiments in section 4(iii) . Note, $\alpha$ follows a similar trend as seen in Figure 12 despite widely different setting. The general recommendation for $\alpha$ is higher the better. Note, the deviation in validation loss after $\alpha = 0.8$ is very small. But this does highlight once potential issue of sensitivity. We counter that by using an adaptive $\alpha$ in our experiments with PROXQUANT and they seem to work well in general. If one wants to use a fixed $\alpha$, we recommend to use 0.9 as we did throughout all our experiments.

### A.5.3. MoXCo + ADAM

Similar behavior is reflected when MoXCo is wrapped with also when used with Ackley function. The figure shows trajectories with same initialization (black circle) and all optimizers run for total of 1000 steps. We see that in both plots Figure 2, 9, only MoXCo variants are able to explore the loss surfer and converge to global minima whereas other optimizers get stuck in unwanted local

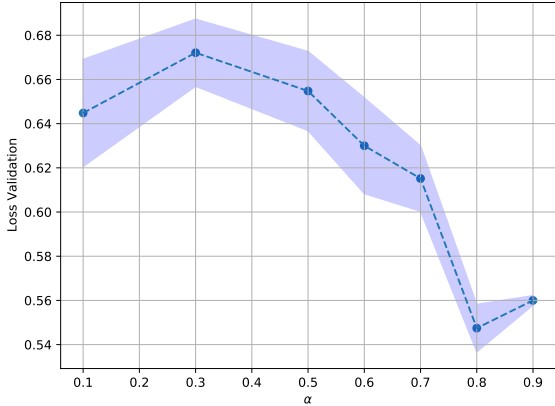

Figure 7: Ablation study for $\alpha$ vs Validation loss for under-parametrized two-layer ReLU network with quadratic loss setting similar to 4(iii)

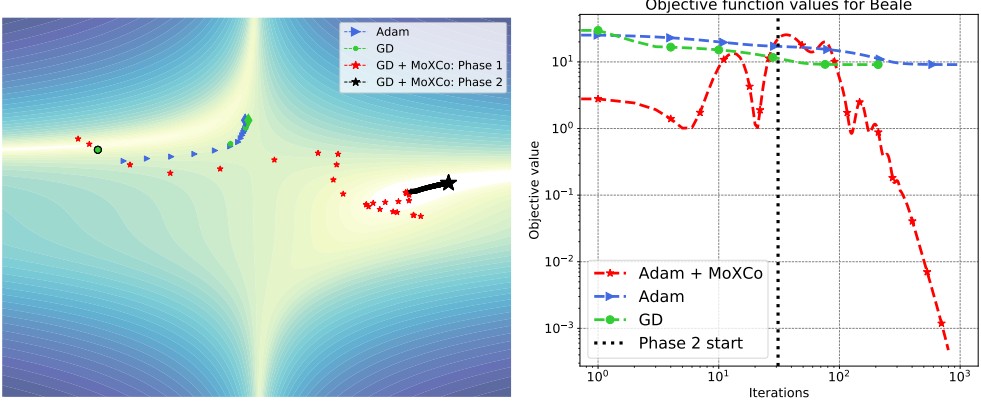

Figure 8: Beale optimization trajectory for same initialization as Figure 2 but this time with ADAM + MoXCo and their vanilla versions. Similar to Fig 2, ADAM + MoXCO is able to escape the local minima. The equations for integrating MoXCo with ADAM are included below. Additionally, we show the learning curves for this case, where vanilla optimizers use the highest stable $\eta = 0.05$ & MoXCo uses $\eta = 0.005$, similar to Fig 2. Despite the smaller rate, the convergence speed of MoXCo is on par with GD, which gets stuck very near its initialization.

stationary points. We used Goodness score of 0.85, and restarting values of $\alpha, \beta = 0.1, 0.1$ for both figures.

Wrapping MoXCo inertial equations with ADAM -

$$
\begin{cases}
\theta_{t+1} = \theta_t + \Delta_t, \\
\hat{m}_t = \dfrac{\beta_1' m_{t-1}}{1 - \beta_1'^t} + \dfrac{(1 - \beta_2')g_t}{1 - \beta_1'^t} \\
\hat{v}_t = \dfrac{\beta_2' v_{t-1}}{1 - \beta_2'^t} + \dfrac{(1 - \beta_2')g_t^2}{1 - \beta_2'^t} \\
\Delta_t = \dfrac{-\gamma \hat{m}_t}{\sqrt{\hat{v}_t} + \epsilon}
\end{cases}
\quad \text{ADAM update rule Kingma and Ba [2014]} \quad (10)
$$

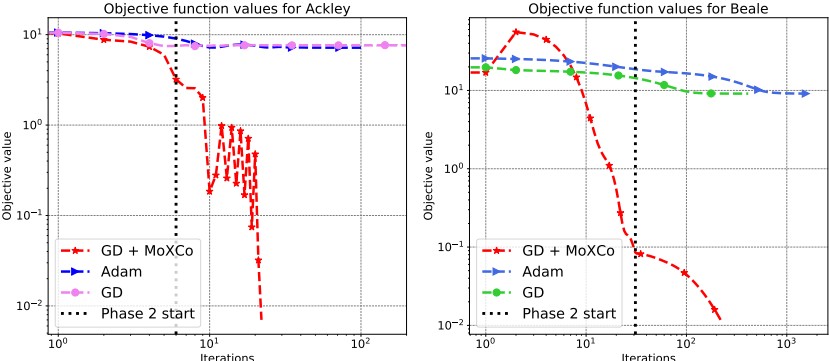

Figure 9: Learning curves for optimization path traced on (a) Ackley corresponding to Figure 1. Notice that MoXCo converges faster than the rest. (b) Beale learning curve wrt to Figure 2, with $\eta = 0.005$ for all optimizers. Note, Phase 2 for (a) starts rather quickly due to large $\alpha$ and step-size.

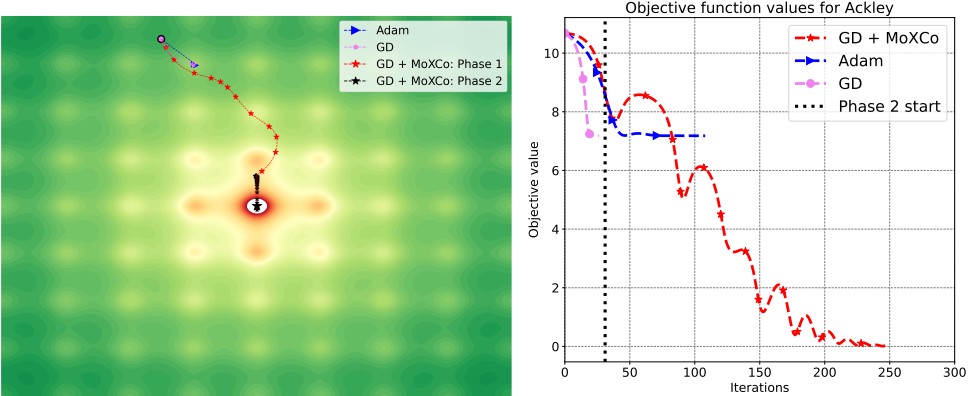

Figure 10: Optimization path traced on (a) Ackley with similar initialization as Fig 4 (b) corresponding learning curve. MoXCo + GD uses $\eta = 0.0005$, rest optimizers use $\eta = 0.05$. ($\eta = 0.01$ also gives similar results). Even with small $\eta$, MoXCo is able to reach global minima.

MoXCo Inertial updates:

$$x_t = \theta_t + \alpha_t(\theta_t - \theta_{t-1})$$
$$y_t = \theta_t + \beta_t(\theta_t - \theta_{t-1})$$
$$\theta_{t+1} = x_t - \eta \nabla L(y_t)$$

Inertial + ADAM:

$$\theta_{t+1} = x_t - \frac{\gamma \hat{m}_t}{\sqrt{\hat{v}_t} + \epsilon}$$
$$m_t = \nabla_\theta L(y_t)$$
$$v_t = (\nabla_\theta L(y_t))^2$$

$\gamma = \eta$ (same as learning rate in Inertial update)

### A.5.4. Additional analysis for non-parametric regression experiments

We provide additional cumulative distribution function (CDF) plots for $m = 2$ in Figure 11, based on 50 different initializations. Throughout these hit rate experiments in the under-parameterized regime, we maintain a consistent configuration of MoXCo: $\alpha, \beta = 0.6, 0.6, \eta = 0.1, \tau = 0.20, \gamma = 0.85$, with restart mechanism using hyper-gradient with $\beta_{\text{hypergradient}} = 1e - 3$. We can replace constant $\beta$

with its randomized version as well. As illustrated in Figure 11, MoXCo achieves lower values across all four metrics, indicating it has converged to a better solution than vanilla GD. We also plot loss values to indicate speed of convergence.

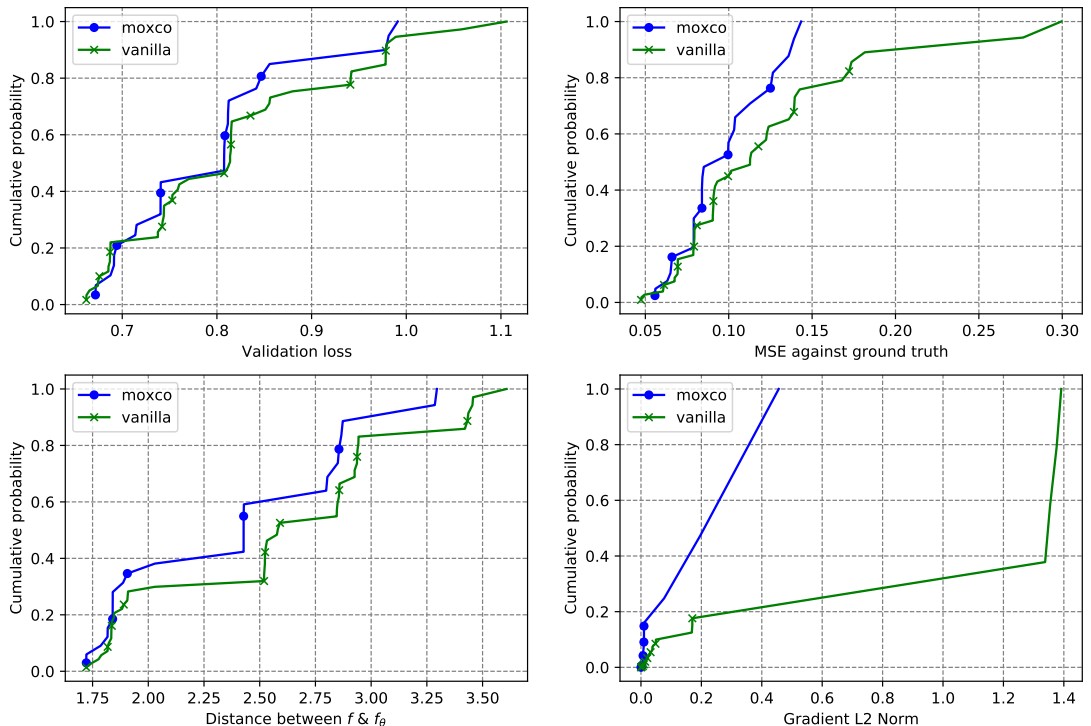

Figure 11: Hit rate analysis for $50$ random initialization when $m = 2$ for under-parameterized two-layer NN optimized using MoXCo + GD. Once again we the evaluate "goodness" of solutions at convergence for these 50 runs using 4 different metrics. Once again, all four metrics are smaller for MoXCo as compared to GD.

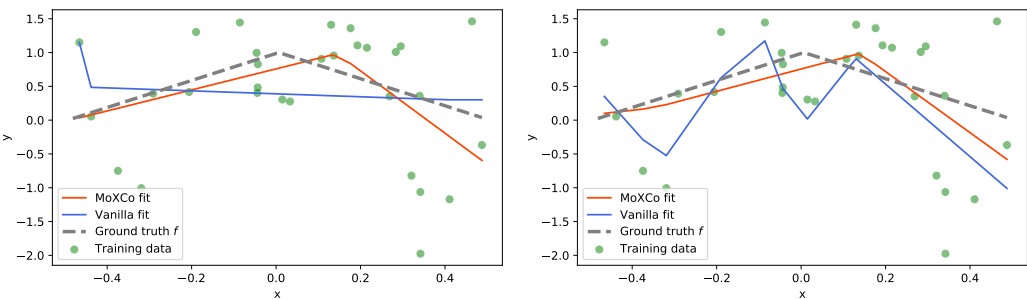

Figure 12: Two different instances of fitted functions randomly initialized with $m = 2$ in the under-parameterized setting. GD fails to learn $f$ in both examples and interpolates, while MoXCo learns the underlying parametric function well.

