# OpenReview forum: "MoXCo: How I learned to stop exploring and love my local minima?"
_CPAL.cc/2025/Proceedings_Track — CPAL 2025 (Proceedings Track) Poster_

### Official Review · Reviewer_mZT2 · 2025-01-07

**Rating:** 6
**Confidence:** 3

**Review:**

**Summary**: This paper presented an optimization method named MoXCo which has two phases: an exploration phase and an exploitation phase. During the exploration phase, the algorithm aims to find a landscape that has some nice properties, satisfied by a goodness score. Once a good landscape is found, the algorithm enters an exploitation phase, where the algorithm will converges to the local minimum.

**Clarity, originality**: This work is clearly presented and is original.

**Pro**: The motivation of the design of this algorithm is clear and is well-motivated. The two-phase design is natural and in terms of which specific algorithm to used in each phase, it opens a door of many choices. In addition, the design of goodness score makes sense and uses idea from landscape flatness.

**Con**: Although the idea of this paper is natural and is solidly executed, I don't feel this paper provides anything that is fundamentally new. On the other hand, I think this paper does provide a way to nicely combining existing methods to enhance optimization.

---

### Official Review · Reviewer_sJCh · 2025-01-09
**Interesting idea but floppy writing**

**Rating:** 5
**Confidence:** 3

**Review:**

The paper proposes a two-stage optimization algorithm that achieves faster convergence and demonstrates superior performance.

**Pros:**
* The proposed two-stage method is interesting and has potential utility for large-scale experiments.
* The experiments conducted are persuasive in demonstrating the effectiveness of the proposed method.

**Cons:**

While the method is interesting, several aspects of the paper are not clearly explained, part of my understanding of the paper is based on guessing. Below I listed some points/questions that I wish the authors to further clarify.
* Starting from Line 216, the discussion appears to detail different terms in Equation (6), but this is not explicitly stated. Furthermore, Line 218 writes "g represents second-order gradient information" with g un-defined.
* Table 1 is not presented clearly, could the authors explain more about the information trying to convey here?
* In Figure 3, the paper uses the notion of Cumulative prob. for comparison of different methods without introducing it. Does it refer to the cumulative sum of values such as MSE or gradient norm for different trials?
* The ResNet experiments are conducted on a binarized version. Is this a deliberate design choice aligned with the proposed method? If so, this should be explicitly justified. Otherwise, why not adhere to the standard ResNet setting?
* Line 168, I don't understand why adding the additional momentum is helpful for non-differentiable objective functions, could the authors elaborate more?

Some typos and suggestions:
* Line 183, "optimizers allowed to can have" should be a typo?
* Line 296, there should be a space between "Metric" and "We".
* All legends/labels used in the figures are too small, it would be better to tweak the figures for better readability. Furthermore, it would be helpful to show a colorbar for Figure 1&2 so that readers can understand lighter color corresponds to smaller loss values.

---

### Official Review · Reviewer_Np8Q · 2025-01-14
**Review for Submission73**

**Rating:** 7
**Confidence:** 3

**Review:**

**Summary**:
This paper introduces the MoXCo (Momentum Exploration and Commit) framework for deep learning optimization. The method addresses the challenges of navigating complex loss landscapes by proposing a two-phase optimization strategy: an aggressive exploration phase and a strategic commitment phase. Key contributions include the introduction of an inertial proximal algorithm to promote exploration and a "Goodness" score heuristic to determine when to transition to convergence. The framework is validated on synthetic functions, neural network training tasks, and model quantization, demonstrating faster convergence and better generalization than traditional methods like SGD and ADAM.

**Strengths**:
1. **Innovative Framework**: The two-phase design (exploration and commitment) is a novel approach to balancing aggressive exploration with efficient convergence.
The use of the "Goodness" score as a heuristic to adaptively switch between phases is both practical and theoretically motivated.

2. **Theoretical and Empirical Rigor**: The paper offers a solid theoretical foundation, including proofs and detailed explanations of the MoXCo framework. Extensive experiments on various tasks and datasets validate the framework’s effectiveness in achieving faster convergence and improved generalization.

3. **Detailed Analysis**: The exploration of curvature properties, Hessian eigenvalues, and gradient norms provides deep insights into why the method outperforms existing optimizers. Visualization of optimization trajectories and metrics enhances understanding of the framework's behavior.

4. **Practical Implementation**: The framework integrates seamlessly with existing optimizers like SGD and ADAM, allowing for easy adoption in practical applications.

**Weaknesses**:
1. **Lack of Real-World Benchmarks**:The method is not tested on large-scale, real-world datasets like ImageNet or NLP tasks, which limits its applicability to practical deep learning challenges.

2. **Generalization to Complex Architectures**: While the method is tested on simple regression tasks and ResNets, its performance on more complex architectures (e.g., transformers, large-scale multi-modal models) remains unexplored.

3. **Simplistic Heuristic Design**: The "Goodness" score, while effective, relies on specific assumptions about the loss landscape. Its adaptability to different types of optimization problems could be better investigated.

---

### Meta-Review · Area_Chair_aV3Q · 2025-02-02

**Recommendation:** Accept (Poster)
**Confidence:** 2

**Metareview:**

Overall, the paper receives mixed yet generally positive reviews (scores: 7, 6, and 5). Reviewers agree the proposed two-phase optimization approach is interesting and well-motivated, providing practical ways to adaptively balance exploration and convergence. The principal concerns relate to clarity (missing details in some explanations), the lack of large-scale benchmarks (e.g., ImageNet), and questions about the fundamental novelty beyond combining known techniques. The authors’ rebuttal addresses some of these points, clarifying the method’s rationale and implementation details. Given the empirical performance, the theoretical grounding, and the detailed responses to reviewer concerns, I lean toward acceptance.

---

### Decision · Program_Chairs · 2025-02-11

Accept (Poster)